# Global biochemical analysis of plasma, serum and whole blood collected using various anticoagulant additives

**Adam D. Kennedy** *, **Lisa Ford, Bryan Wittmann, Jesse Conner, Jacob Wulff, Matthew Mitchell, Anne M. Evans, Douglas R. Toal**

Metabolon, Morrisville, North Carolina, United States of America

* adamillini4@gmail.com

**Data Availability Statement:** All relevant data are within the paper and its Supporting information files.

## Abstract

### Introduction

Analysis of blood for the evaluation of clinically relevant biomarkers requires precise collection and sample handling by phlebotomists and laboratory staff. An important consideration for the clinical application of metabolomics are the different anticoagulants utilized for sample collection. Most studies that have characterized differences in metabolite levels in various blood collection tubes have focused on single analytes. We define analyte levels on a global metabolomics platform following blood sampling using five different, but commonly used, clinical laboratory blood collection tubes (i.e., plasma anticoagulated with either EDTA, lithium heparin or sodium citrate, along with no additive (serum), and EDTA anticoagulated whole blood).

### Methods

Using an untargeted metabolomics platform we analyzed five sample types after all had been collected and stored at -80˚C. The biochemical composition was determined and differences between the samples established using matched-pair t-tests.

### Results

We identified 1,117 biochemicals across all samples and detected a mean of 1,036 in the sample groups. Compared to the levels of metabolites in EDTA plasma, the number of biochemicals present at statistically significant different levels (p<0.05) ranged from 452 (serum) to 917 (whole blood). Several metabolites linked to screening assays for rare diseases including acylcarnitines, bilirubin and heme metabolites, nucleosides, and redox balance metabolites varied significantly across the sample collection types.

### Conclusions

Our study highlights the widespread effects and importance of using consistent additives for assessing small molecule levels in clinical metabolomics. The biochemistry that occurs

**Funding:** The funder provided support in the form of salaries for authors [AK, LF, BW, JC, JW, MM, AE, DT], but did not have any additional role in the study design, data collection and analysis, decision to publish, or preparation of the manuscript. The specific roles of these authors are articulated in the 'author contributions' section.

**Competing interests:** AK, LF, BW, JC, JW, MM, AE, DT are employees of Metabolon, Inc. and, as such, have affiliations with or financial involvement with Metabolon, Inc. The authors have no other relevant affiliations or financial involvement with any organization or entity with a financial interest in or financial conflict with the subject matter or materials discussed in the manuscript apart from those disclosed. This does not alter our adherence to PLOS ONE policies on sharing data and materials.

**Abbreviations:** EDTA, Ethylenediaminetetraacetic acid; LC-MS, liquid chromatography tandemly linked to mass spectrometry.

during the blood collection process creates a reproducible signal that can identify specimens collected with different anticoagulants in metabolomic studies.

## Impact statement

In this manuscript, normal/healthy donors had peripheral blood collected using multiple anticoagulants as well as serum during a fasted blood draw. Global metabolomics is a new technology being utilized to draw clinical conclusions and we interrogated the effects of different anticoagulants on the levels of biochemicals from each of the donors. Characterizing the effects of the anticoagulants on biochemical levels will help researchers leverage the information using global metabolomics in order to make conclusions regarding important disease biomarkers.

## Introduction

The analysis of large cohorts of samples has become an increasingly common approach to analyze disease signatures as well as identify new biomarkers of disease [1–4]. The ability to analyze large numbers of samples is no longer rate limiting, rather the limiting aspect of metabolomic studies is ensuring a strong study design that allows for the interrogation of thousands of samples and requires that the study samples are collected using the same blood collection device. Metabolomics has been a leading technology in small molecule biomarker discovery and complements other technologies such as genomics [5–8]. Many biobanks exist that house disease-specific sample cohorts for metabolomic discovery. Inasmuch as blood metabolite levels in cohort samples are influenced by the method of collection, it is important to consider all aspects of sample collection, especially selection of the appropriate collection tube. Therefore, collecting samples in a consistent manner and choosing the correct sample type are paramount to maximize data acquisition to ensure accurate conclusions made from large population cohort studies [9–13]. The analysis of these studies, when performed optimally, can result in the identification of new and more informative biomarkers of disease and treatment of disease.

For many current diagnostics, biomarkers are well known. These include glucose and hemoglobin A1c for monitoring glycemic index and cholesterol, triglycerides, and high- and low-density lipoproteins for monitoring risk of developing cardiovascular disease. Strict analytical and clinical validation studies reveal that these are strong indicators for these conditions affected by dysregulated glucose and lipid metabolism, specifically. However, as medical care has progressed, there is an increasing need to expand biomarker panels in order to identify early and lead indicators of disease as well as markers that inform on the effectiveness of therapeutic interventions [14].

Clinical assays provide diagnostic insight for several disease states, yet opportunities exist to identify new biomarkers that improve upon current clinical assay performance and expand diagnostic reach to diseases that are underserved by the routine repertoire of clinical biomarkers [15]. The expansion of diagnostic assays using new biomarkers, whether small molecules or proteins, can give increased resolution to both common diseases as well as rare diseases [16]. This can be accomplished through targeted panels of compounds or through clinical metabolomics, the analysis of the comprehensive metabolome of a sample. Recently, validation of a metabolomics platform has shown that precise analysis of hundreds of molecules can be

accomplished simultaneously in order to screen for clinical signatures in human plasma [17]. This validation focused on the accuracy, intra-day and inter-day precision assessment of over 250 molecules detected on this platform. The median precision for these molecules was <6%. This same validated platform was utilized for this study. For example, glomerular filtration rate (GFR) is routinely measured using serum creatinine levels to calculate estimated GFR (eGFR). This measurement can have as much as 30% error compared to measured GFR. New diagnostic biomarkers have been identified that can assess kidney function much more accurately but knowing in which matrix to measure these markers is crucial [18, 19]. As diagnostic panels are expanded, rapid assays are needed in order assess a patient's condition and allow clinicians to make rapid diagnostic conclusions. For example, identifying biomarkers of disease for rare disorders and inborn errors of metabolism requires the analysis of molecules outside of typical panels run in newborn screening [20–22]. The analysis of novel biomarker signatures has increased the utility of EDTA plasma, urine and cerebrospinal fluid through the identification of new compounds that can be added to screening assays [23–32]. In addition to identifying biomarkers of disease, researchers and clinicians need to identify and track biomarkers showing that treatment of a disease is effective as well as markers that could indicate if a toxicological response is occurring in a patient while taking a therapeutic intervention.

In this study, we sought to characterize the complete metabolome of five clinically relevant blood matrices: plasma anticoagulated with either EDTA, sodium citrate, or lithium heparin, serum with no preservative, and EDTA anticoagulated whole blood. The information obtained by the direct comparison of each of the biochemicals in all five of these sample types would allow for a greater understanding of how these biochemicals behave in samples obtained from normal/healthy individuals. We examined this dataset to identify the differences in levels of biochemicals in match-paired samples associated with the use of different anti-coagulants commonly used in the clinical laboratory. Although differences in the levels of some of these biochemicals are known, to our knowledge this is the first comprehensive analysis of the metabolome directly comparing these different sample types. Results of these analyses demonstrate that accurate identification of samples exposed to different anticoagulants will be needed in order to perform accurate and precise clinical metabolomic assays.

## Materials and methods

### Sample collection

All procedures were performed in accordance with the ethical standards of the U.S. Department of Health and Human Services and were approved by an Institutional Review Board (IRB), (Aspire IRB, Santee, CA). Specimens used in metabolomic testing were collected from donors through informed consent at Metabolon's clinical laboratory (Morrisville, NC).

Whole blood, serum and plasma, collected in containers with different anticoagulants (lithium heparin, potassium EDTA, and sodium citrate for plasma and no anticoagulant for serum), from twenty-seven (27) subjects, resulted in five sample types and 135 total samples. Samples used for serum analysis were coagulated for 30 minutes. All samples, except whole blood samples, were centrifuged to collect the respective specimens. All individuals fasted for 8 hours prior to blood collection. Multiple aliquots were stored and frozen for analysis. All samples were stored at -80˚C for 1 month prior to metabolomic testing. The average age for the subjects was 37 years (range from 24 to 68 years of age, median of 34 years of age) and 44% female (n = 12). None of the subjects in the study were pregnant at the time of their blood draw. It was a requirement that no subjects were pregnant at the time of their draw. Although we are aware that menstrual status of subjects can affect the biochemical profile of a donor, the analyses performed in our statistical analyses were centered around matched-pair t-tests to

identify differences from sample type to sample type within an individual. The menstrual cycle status of the subjects in the study is important, but not a central factor in the analysis. The metabolites that change with menstrual status would contribute to the range of detection in the study.

Some of the patients were taking prescription medications and any medications that fit the profile of a small molecule that were detected in the study are listed in S1 Table (Xenobiotics superfamily). Medications can alter the biochemical profiles of patients but the analyses performed in our statistical analyses were centered around matched-pair t-tests in order to identify differences from sample type to sample type within an individual.

## Metabolomic analysis

**Sample preparation.** Metabolomics was performed as described previously [33, 34]. One hundred microliters of sample was used for each analysis. Small molecules were extracted in an 80% methanol solution containing recovery standards [35]. Proteins were precipitated from 100 $\mu$L of human plasma/serum/whole blood with methanol using an automated liquid handler (Hamilton LabStar). The methanol contained internal standards specific for each chromatographic method, which permitted the monitoring of extraction efficiency. The precipitated extract was split into four aliquots and dried under nitrogen and then in vacuum. The samples were then reconstituted in the appropriate buffers for each chromatographic method and then data acquired on the LC/MS instruments. During the data analysis, process balnks are run throughout the data acquisition to identify molecules that are present in tubes and solvents. Any compound not present 3× above water blank levels are removed.

**LC/MS/MS$^n$ analyses.** All methods utilized a Waters ACQUITY ultra-performance liquid chromatography (UPLC) and a Thermo Scientific Q-Exactive high resolution/accurate mass spectrometer interfaced with a heated electrospray ionization (HESI-II) source and Q-Exactive mass analyzer operated at 35,000 mass resolution [17, 34]. The dried sample extract aliquots were reconstituted in solvents compatible to each of the four LC-MS/MS methods with a series of isotopically labeled standards at fixed concentrations to monitor injection and chromatographic consistency and to align chromatograms (S1 Fig). Separate aliquots were analyzed by two reverse phase positive ion methods, one reverse phase negative ion method, and one hydrophilic interaction liquid chromatographic method [34]. Raw data files were archived, and data extracted as described below.

Metabolites were identified by matching the ion chromatographic retention index, accurate mass, and mass spectral fragmentation signatures with a reference library consisting of over 4,000 entries created from authentic standard metabolites under the identical analytical procedure as the experimental samples [33]. Identification of compounds was based on the match of its retention time, parent ion accurate mass, and MS/MS fragmentation spectrum to an authentic standard representing Tier 1 identification [36]. Compounds marked by an asterisk were identified based on parent ion accurate mass and MS/MS fragmentation mass spectral data without a reference standard (i.e., Tier 2 identification). Molecules not detected in a sample were below the limit of detection.

This project was run as part of a larger project and all samples were in multiple batches across each arm of the platform, but samples from a single donor were analyzed in the same batch for each of the MS/MS instruments. All samples were run as a single batch and randomized into four groups of 33 samples each. Samples from a single donor were analyzed across multiple batches for each of the MS/MS instruments. A single aliquot of each sample from each donor was run in the study. A pooled sample from >100 donors outside of this study was run as technical replicates throughout the study to monitor process variability and quality.

Within each sample batch, 2 process blanks and 6 quality control samples are run and analyzed. For this analysis, there were four sample batches meaning 8 process blanks and 24 quality control samples were run in total. The median relative standard deviation (RSD) was calculated for all standards and endogenous biochemicals using median scaled values. The median RSDs for the internal standards and endogenous biochemicals were 4% and 11%, respectively.

## Data analysis and statistics

Raw ion area under the curve (AUC) values were batch normalized by dividing each batch by the median scaled to generate scaled AUCs followed by imputation of any missing values with a value based on the minimum detected value, and, finally, were natural log-transformed on a per biochemical basis. The log transformed data were utilized for significance testing. Mean differences were tested using matched-pair t-tests and multiple comparisons were accounted for using a false discovery rate (FDR) approach estimated by the q-value method of Tibshirani and Storey [37, 38]. Correlations were calculated by the Pearson method. Correlations for individual metabolites used untransformed values. Hierarchical clustering was performed using the log-transformed, normalized, imputed data with complete clustering and a Euclidean distance metric.

## Results

### Metabolomic profile of blood samples collected using different anticoagulants

We identified 1,177 biochemicals across 135 samples, representing all donors and all sample types, that matched biochemical entries in the library (S1 Table). The mean number of biochemicals detected per sample was 1,036 (median 1,039) with a range of 954 to 1,077 biochemicals. Serum contained the most biochemicals with an average of 1,071 (range 1,016–1,177) followed by EDTA plasma (mean 1,049, range 997–1,105), heparinized plasma (mean 1,048, range 994–1,087), citrate plasma (mean 1,013, range 961–1,078), and whole blood (mean 998, range 954 to 1,047). Five hundred three (503) out of 1,177 biochemicals (45% of the total) were detected in every sample, 675 biochemicals (61% of total) were detected in ≥ 90% of the samples, and 775 biochemicals (70% of total) were detected in ≥ 80% of the samples.

Two unsupervised analyses showed the global relationship between the sample types (Fig 1 and S2 Fig). Principal component analysis revealed a strong separation between the whole blood and serum/plasma samples (Fig 1). Within the plasma and serum samples, EDTA plasma and citrate plasma showed a slight segregation from the lithium heparin plasma and serum. There was a more apparent separation of the EDTA plasma and citrate plasma samples compared to the lithium heparin and serum samples. The lithium heparin and serum samples showed very similar profiles and did not show any apparent separation. Hierarchical clustering showed three major clades (S2 Fig). The first was a strong segregation of the whole blood samples from all other sample collection methods and the second segregated the citrate and EDTA plasma samples. The third clade contained the heparin plasma and serum samples which did not segregate from one another indicating similar biochemical profiles within the samples.

EDTA chelates divalent metal cations; molecules necessary for the activation of several enzymes expressed in whole blood, including the clotting cascade. Given that EDTA plasma is used as a central diagnostic sample and that the chelation of these metal cations inhibits nearly all protein activity, we used the EDTA plasma sample group as the central comparator for the study. Specifically, matched-pair t-tests comparing each blood collection type to EDTA plasma

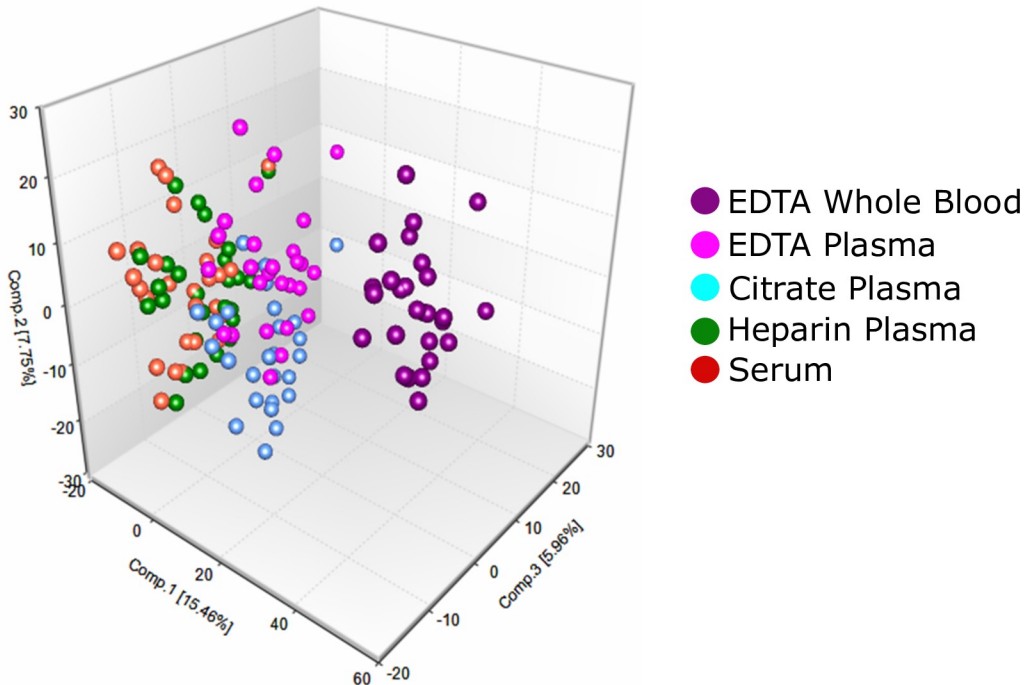

**Fig 1. Unsupervised statistical analysis for all samples.** Principal component analysis of the five sample types. Dark Purple indicates the EDTA anticoagulated whole blood samples; light purple—EDTA plasma samples, light blue—citrate plasma; green—heparinized plasma, and orange—serum samples.

revealed numerous biochemical differences (Table 1). The largest number of differences was between EDTA plasma and whole blood and citrate plasma with 917 and 807 statistically significant differences, respectively. Heparin plasma and serum had 507 and 452 differences respectively and nearly all of them were of the same magnitude and direction when compared to EDTA plasma (S1 and S2 Tables). The ranges of fold changes varied among the sample types and are outlined in S2 Table. The largest differences were attributable to biochemicals detected in specific sample types (e.g. fibrinogen peptides in serum, glutathione in whole blood) and may explain why serum and lithium heparin plasma were separated strongly from EDTA-plasma.

## Select biochemical families altered by the sample type

Clinical diagnosis of specific conditions such as jaundice, liver function, disorders associated with metabolic disease, and assessment of hormones/neurotransmitter metabolites in the

**Table 1. Number of significantly altered biochemicals in blood collected in different tube collection types.**

| | Serum vs. EDTA plasma | Heparin plasma vs. EDTA plasma | Citrate plasma vs. EDTA plasma | Whole blood vs. EDTA plasma |
|---|---|---|---|---|
| Biochemical levels increased versus EDTA Plasma | 243 | 178 | 72 | 270 |
| Biochemical levels decreased versus EDTA plasma | 209 | 329 | 735 | 647 |
| Total number of significantly altered biochemicals | 452 | 507 | 807 | 917 |

bloodstream require the collection of specific sample types. Many of these biomarkers are utilized for routine diagnostics in pediatric patients and newborn screening. In the following section, we assessed the levels of several biochemicals utilized for the diagnosis of these conditions.

The neurotransmitter, serotonin, is produced in the central nervous system and can be measured in the bloodstream, with 90% found in platelets, which lyse during the blood clotting process during serum collection. Consequently, in our evaluation, serotonin was present at 17× in serum and 4× in whole blood compared to EDTA plasma (Fig 2A, S1 Table). Conversely, bilirubin functions as an antioxidant in the blood stream and is the most abundant antioxidant in the plasma and serum fraction of whole blood. We identified heme, five bilirubin metabolites/isoforms as well as seven bilirubin degradation products in this study (Table 2). Heme and biliverdin are erythrocyte-associated compounds that are catabolized to bilirubin and, as expected, we observed a 121× and 3× increase, respectively, in these two compounds in whole blood compared to EDTA plasma (Table 2). Bilirubin, on the other hand, is present in serum and plasma and nearly absent in whole blood samples (Fig 2B).

The family of molecules categorized as acylcarnitines are monitored in patients diagnosed with or suspected of having rare diseases associated with lipid metabolism and lipid transport. Citrate plasma and whole blood showed the most diverse acylcarnitine signature as compared to EDTA plasma (Table 3). The long-chain acylcarnitines with 14 to 18 carbons showed significant elevations in whole blood versus EDTA plasma whereas the same molecules showed ~25% lower levels in citrate plasma versus EDTA plasma. Acylcarnitines decreased by an average of 24% across 27 different molecules in citrate plasma when samples were collected using EDTA as an anticoagulant.

Nucleotides are ubiquitous metabolic compounds expressed in all tissues and play significant roles in cellular signaling [39] in addition to serving as molecular components for DNA synthesis. Specifically, purine metabolite adenosine plays a significant role in regulating cardiac function. Adenosine was measured in all sample types but present at 13x higher levels in serum versus EDTA plasma and 2× higher levels in whole blood versus EDTA plasma (Fig 2C). The nucleosides guanosine and inosine have been linked to neuropathologies and neuroprotective roles in studies examining Parkinson's disease and depression. Both nucleosides showed significantly elevated levels in EDTA plasma versus serum 167× and 51×, respectively, as well as elevated levels in heparinized plasma 16× and 4×, respectively (Fig 2D and 2E).

## Evaluation of biomarkers associated with kidney function and sample types

Small molecules associated with kidney function have been a recent expansion in the understanding of kidney function as well as obtaining a more accurate assessment of GFR. The panel of biomarkers consisting of C-glycosyltryptophan, pseudouridine, N-acetylthreonine, phenylacetylglutamine and creatinine can be leveraged to assess kidney function (43). Although the fold-of-change was relatively small between the sample types, several of the comparisons revealed statistically significant differences (Fig 3). The standard collection in the United States for assessing kidney function is through serum collection, but in several countries throughout Europe, heparinized plasma is the sample of choice. These five metabolites showed strong correlations when the levels of the biochemicals were compared between serum and lithium heparinized plasma (Fig 4). Furthermore, correlation between serum and lithium heparin plasma versus EDTA plasma for these kidney function biomarkers were high (S1 Table).

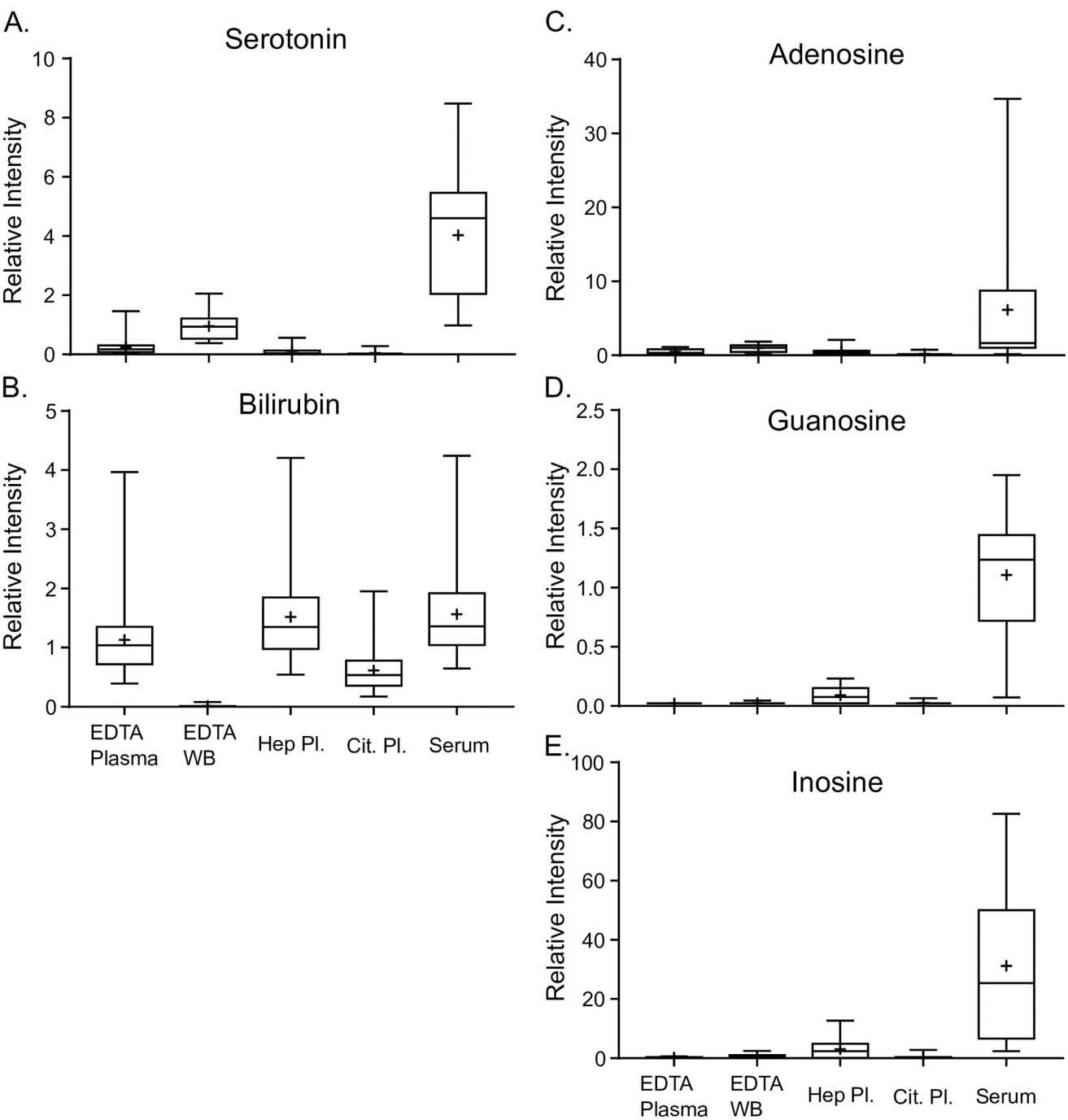

**Fig 2. Differences in levels for several clinically relevant biochemicals.** (A)Box plot for serotonin. The boxes outline the $2^{nd}$ and $3^{rd}$ quartile (middle 50%) of the data for each sample type. (B) Box plot for bilirubin. (C), (D), and (E) Box plots for Adenosine, Guanosine, and Inosine, respectively. The error bars on the graph represent $1.5 \times$ IQR (interquartile range for that metabolite within the sample type. The line within the box represents the median of the data and the "+" symbol represents the mean of the data set.

## Distinctive biochemicals that identify the sample type

The analysis detected six fibrinogen peptides in serum, a result of the clotting process, that were absent in all other sample types. The levels of citrate in sodium citrate plasma samples

**Table 2. Heme, bilirubin, and bilirubin degradation products in whole blood, plasma and serum represented as fold-of-change compared to levels in EDTA plasma.**

| Heme and bilirubin analytes | Serum | Heparinized plasma | Citrate plasma | Whole blood |
|---|---|---|---|---|
| Heme | **0.36** | **0.27** | **0.27** | **121.12** |
| bilirubin (Z,Z) | **1.06** | 1.03 | **0.83** | **0.05** |
| bilirubin (E,E)* | **1.39** | **1.35** | **0.54** | **0.01** |
| bilirubin (E,Z or Z,E)* | 1.08 | 1.04 | 0.96 | **0.27** |
| Biliverdin | **0.79** | **0.7** | **0.51** | **3.17** |
| bilirubin degradation product, $C_{16}H_{18}N_2O_5$ (1)* | **0.73** | **0.63** | **0.38** | **0.74** |
| bilirubin degradation product, $C_{16}H_{18}N_2O_5$ (2)* | **0.67** | **0.59** | **0.35** | **0.41** |
| bilirubin degradation product, $C_{17}H_{18}N_2O_4$ (1)* | **1.2** | 1.07 | **0.51** | 1.16 |
| bilirubin degradation product, $C_{17}H_{18}N_2O_4$ (2)* | 1.14 | 0.96 | **0.55** | **0.87** |
| bilirubin degradation product, $C_{17}H_{18}N_2O_4$ (3)* | 1.1 | 0.93 | **0.54** | **0.65** |
| bilirubin degradation product, $C_{17}H_{20}N_2O_5$ (1)* | 0.86 | **0.62** | **0.3** | 0.94 |
| bilirubin degradation product, $C_{17}H_{20}N_2O_5$ (2)* | 0.92 | **0.68** | **0.33** | 0.86 |
| bilirubin degradation product, $C_{16}H_{18}N_2O_5$ (3)* | **0.65** | **0.6** | **0.41** | 0.87 |
| bilirubin degradation product, $C_{16}H_{18}N_2O_5$ (4)* | **0.66** | **0.6** | **0.34** | **0.47** |

Values in bold font refer to statistically significant differences per matched-pair t-test analysis.

* indicates a compound that has not been officially confirmed based on a standard, but we are confident in its identity.

were 34× higher than EDTA plasma samples. Meanwhile, EDTA and iminodiacetate, a degradation product of EDTA, were detected in EDTA plasma and EDTA anticoagulated whole blood and none of the other sample types (S1 Table).

## Discussion

The choice of sample type for metabolomic studies influence the number and levels of compounds detected in a population. This is significant in clinical metabolomic approaches since the reference range for each compound is dependent on the sample type. The identification of novel biomarkers of disease and the ability to monitor therapeutic intervention of disease will be critical elements that need to occur for novel therapeutics currently under development and for those that will be in development in the future. The biochemical profile of whole blood, serum and plasma is a complex composition of over one thousand small molecules representing multiple biochemical families. In this study, we compared the levels of each compound detected in five different sample types (S1 Table) as well as the relative ranges of these biochemicals (S2 Table). Both aspects are important for the consideration of a biochemical in a diagnostic assay given that large dynamic ranges may be necessary for identifying healthy patients versus those with a disease as well as determining if treatment of a disease is efficacious. Changes in clinical assay performance caused by blood collection tube additives are an important but often overlooked variable that contributes to the performance of a compound or panel of compounds in an assay. For instance, EDTA binds to divalent metal cations ($Ca^{2+}$ and $Mg^{2+}$) required for enzyme cofactors used for immunoassay reagents such as alkaline phosphatase [40], heparin binds with electrolytes and changes the concentrations of bound and free ions which can influence clinical assays [41], and sodium citrate inhibits aspartate aminotransferase and alkaline phosphatase by chelation of cations [42]. The full effects of the different anticoagulants have been reviewed by in several articles as outlined in the following references [42–47]. Lithium heparin and serum have differences between the two sample types driven by the catabolism of fibrinogen and the lysis of cellular elements during blood clotting.

**Table 3. Fold-of-change in acylcarnitine levels in the blood specimens versus EDTA plasma.**

| Lipid Subfamily | Acylcarnitine | Serum | Heparinized plasma | Citrate plasma | Whole blood |
|---|---|---|---|---|---|
| Fatty Acid Metabolism (Acyl Carnitine, Short Chain) | acetylcarnitine (C2) | **0.61** | **0.63** | **0.78** | **2.71** |
| Fatty Acid Metabolism (also BCAA Metabolism) | butyrylcarnitine (C4) | **0.85** | **0.83** | **0.78** | **0.74** |
| | propionylcarnitine (C3) | **0.76** | **0.76** | **0.8** | **5.57** |
| Fatty Acid Metabolism (Acyl Carnitine, Medium Chain) | hexanoylcarnitine (C6) | **0.89** | **0.87** | **0.8** | **0.84** |
| | octanoylcarnitine (C8) | **0.93** | **0.89** | **0.78** | **0.52** |
| | cis-3,4-methyleneheptanoylcarnitine | **0.88** | **0.87** | **0.78** | **0.61** |
| | nonanoylcarnitine (C9) | **0.88** | **0.91** | **0.78** | **0.47** |
| | decanoylcarnitine (C10) | 0.96 | **0.9** | **0.77** | **0.53** |
| | laurylcarnitine (C12) | 0.96 | **0.93** | **0.78** | **0.59** |
| Fatty Acid Metabolism (Acyl Carnitine, Long Chain Saturated) | myristoylcarnitine (C14) | 0.97 | **0.93** | **0.75** | **3.03** |
| | palmitoylcarnitine (C16) | **0.96** | **0.92** | **0.75** | **9.7** |
| | margaroylcarnitine (C17)* | 0.97 | 0.96 | **0.77** | **11.34** |
| | stearoylcarnitine (C18) | 0.98 | 0.97 | **0.76** | **16.64** |
| | arachidoylcarnitine (C20)* | **0.89** | **0.9** | **0.79** | **4.47** |
| | behenoylcarnitine (C22)* | **0.79** | **0.69** | **0.66** | **1.98** |
| | lignoceroylcarnitine (C24)* | **0.86** | **0.85** | **0.76** | **1.14** |
| | cerotoylcarnitine (C26)* | **0.86** | **0.86** | **0.77** | **1.14** |
| Fatty Acid Metabolism (Acyl Carnitine, Monounsaturated) | 3-decenoylcarnitine | **0.68** | **0.44** | 0.75 | 0.66 |
| | cis-4-decenoylcarnitine (C10:1) | **0.92** | **0.92** | **0.77** | **0.49** |
| | undecenoylcarnitine (C11:1) | **0.94** | **0.9** | **0.77** | **0.55** |
| | 5-dodecenoylcarnitine (C12:1) | 0.99 | 1 | **0.76** | **0.52** |
| | myristoleoylcarnitine (C14:1)* | **0.96** | **0.9** | **0.77** | **0.6** |
| | palmitoleoylcarnitine (C16:1)* | **0.95** | **0.91** | **0.77** | **2.95** |
| | oleoylcarnitine (C18:1) | 0.99 | **0.89** | **0.61** | **17.38** |
| | eicosenoylcarnitine (C20:1)* | **0.83** | **0.87** | **0.75** | **8.46** |
| | ximenoylcarnitine (C26:1)* | **0.87** | **0.9** | **0.76** | **1.06** |
| Fatty Acid Metabolism (Acyl Carnitine, Polyunsaturated) | linoleoylcarnitine (C18:2)* | 0.98 | **0.93** | **0.75** | **8.37** |
| | dihomo-linoleoylcarnitine (C20:2)* | **0.91** | **0.88** | **0.68** | **11.15** |
| | arachidonoylcarnitine (C20:4) | 0.94 | **0.91** | **0.69** | **5.83** |
| | dihomo-linolenoylcarnitine (C20:3n3 or 6)* | 0.96 | **0.91** | **0.71** | **7.3** |
| Fatty Acid Metabolism (Acyl Carnitine, Dicarboxylate) | adipoylcarnitine (C6-DC) | **0.84** | **0.85** | **0.89** | **0.53** |
| | pimeloylcarnitine/3-methyladipoylcarnitine (C7-DC) | 0.93 | 0.97 | 0.9 | **0.41** |
| | suberoylcarnitine (C8-DC) | 0.96 | 0.95 | **3.35** | **0.58** |
| | octadecanedioylcarnitine (C18-DC)* | 0.97 | **0.92** | **0.74** | **0.5** |
| | octadecenedioylcarnitine (C18:1-DC)* | **0.89** | **0.93** | **0.8** | **0.52** |
| Fatty Acid Metabolism (Acyl Carnitine, Hydroxy) | (R)-3-hydroxybutyrylcarnitine | **0.76** | **0.76** | **0.75** | **3.23** |
| | (S)-3-hydroxybutyrylcarnitine | **0.57** | **0.63** | 1.03 | **5.63** |
| | 3-hydroxyoctanoylcarnitine (1) | **0.58** | **0.59** | **0.77** | 0.99 |
| | 3-hydroxyoctanoylcarnitine (2) | **0.75** | **0.76** | **1.09** | **0.7** |
| | 3-hydroxydecanoylcarnitine | **0.69** | **0.65** | **0.74** | **0.63** |

Values in bold font refer to statistically significant differences per matched-pair t-test analysis.

*—molecules whose structures have been elucidated through $MS^n$ analysis but no purified standard has been analyzed for confirmation.

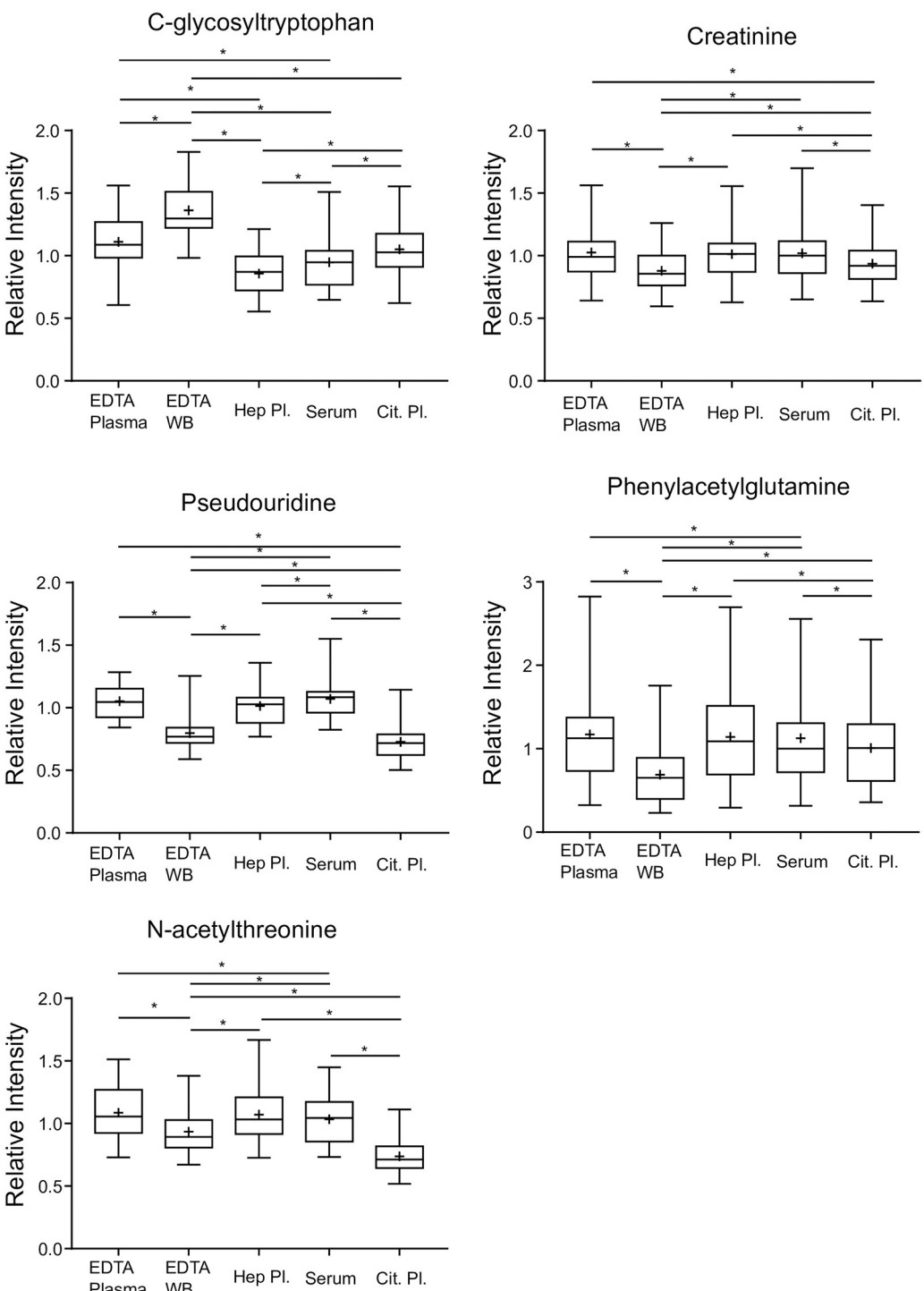

**Fig 3. Box plots for biomarkers of kidney function.** Box plots for C-glycosyltryptophan, pseudouridine, N-acetylthreonine, creatinine, and phenylacetylglutamine are outlined. The boxes outline the 2nd and 3rd quartile (middle 50%) of the data for each sample type. The error bars on the graph represent 1.5 × IQR (interquartile range for that metabolite within the sample type. The line within the box represents the median of the data and the "+" symbol represents the mean of the data set. An "∗" indicates a $p < 0.05$ for the matched-pair comparison between the indicated sample types.

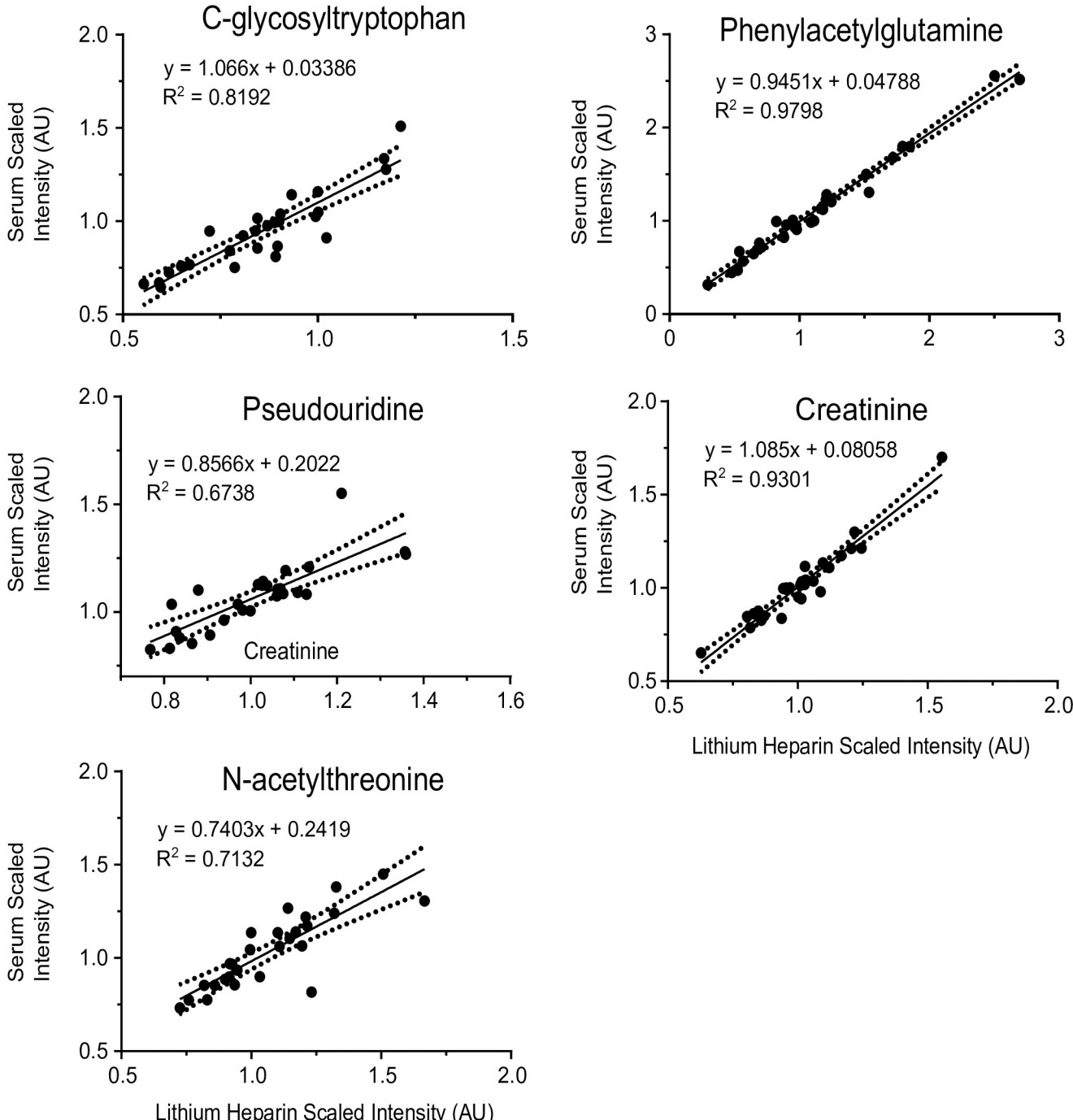

**Fig 4. Linear correlation plots for biomarkers of kidney function.** Scatter plots for C-glycosyltryptophan, pseudouridine, N-acetylthreonine, creatinine, and phenylacetylglutamine are outlined. The values for each metabolite were plotted for the serum and lithium heparin samples for all 27 donors in the study. The equation and $R^2$ values for the best fit lines are outlined within each graph. The dotted lines indicate the 95% confidence interval for each linear fit.

EDTA can cause physiologic changes to neutrophils in whole blood as well as morphological changes in platelets. In both cell types, degranulation is inhibited during the anticoagulation process. Unlike heparin and EDTA, anticoagulation with citrate is preferred for monitoring routine coagulation testing. As with EDTA, citrate can cause morphological changes in white

cells and platelets. As clinical metabolomics gains more traction in medical diagnostics, identifying the most optimal sample type is particularly relevant since hundreds of metabolites are measured, and changes in tube additives can have significant effects on test outcome.

Although serum is commonly used in clinical laboratory testing, clinical metabolomics requires rapid processing times to preserve metabolite levels and therefore EDTA plasma is the preferred sample type. To fully catalog the effects of different blood collection tubes on clinical metabolomic applications, we measured the metabolic profile of human blood collected in five different blood collection tubes from 27 subjects. The results demonstrated that while 90% of the 1,117 analytes measured were found in most of the samples, significant differences existed in analyte levels across the blood collection tube types (Table 1 and S1 and S2 Tables). PCA and hierarchical clustering of the data shows that sodium heparin plasma and serum segregated together and were discriminated from EDTA plasma and sodium citrate plasma (Fig 1 and S2 Fig). Whole blood, as expected, segregated from plasma and serum due to the contribution of cellular components. Furthermore, we detected significant differences in distinct metabolic pathways and highlighted three examples: 1) neurological function as defined by serotonin levels (Fig 2A), 2) redox balance and liver metabolism (Fig 2B) and 3) molecules linked in inter- and extracellular signaling cascades (Fig 2C–2E). These results define the contribution of blood collection tube additives to analyte measurements in metabolomic applications and emphasize the need for careful consideration when selecting the correct blood collection tube for analysis.

In a recent publication [17], we described a relative-quantitation approach to clinical metabolomics where individual analyte measurements in a patient EDTA plasma sample are normalized against replicate EDTA plasma samples in the batch and then compared to an identically normalized reference population to produce analyte z-scores. Since analyte measurements in a patient sample are compared to identically prepared samples of a large reference population, and since we show here that analyte levels are significantly affected by tube additives, it is critical that the patient sample is collected in the correct tube (e.g., potassium EDTA plasma for clinical samples) and processed immediately. To quench effects of ongoing metabolism following sample collection, we require that plasma is separated from the cell component within one hour of collection and that the plasma is aliquoted in a separate tube and immediately frozen at -80˚C [9]. One of the many benefits of measuring hundreds of analytes in a patient sample via clinical metabolomics is that it allows surveillance of biomarkers that predict pre-analytical processing errors. Jain et al., [9] used erythrocyte metabolism markers to accurately identify samples with delayed plasma collection with a diagnostic accuracy of 100%. We add to the repertoire of pre-analytical processing error biomarkers by demonstrating that fibrinogen peptides are only found in serum samples, EDTA and iminodiacetate are only found in potassium EDTA plasma and that high levels of citrate can be used to distinguish samples collected in sodium citrate plasma tubes. Unlike routine clinical laboratory testing where patient samples are received by the laboratory in colored tube tops that distinguish tube type, samples that are delivered to our clinical metabolomics laboratory have been aliquoted and frozen in cryovials and it is therefore not possible to confirm correct tube type. Nevertheless, we have implemented routine quality control steps to assess pre-analytical processing error biomarkers in each patient sample to confirm correct sample type (e.g., presence of EDTA and iminodiacetate) and reject samples that do not meet acceptable criteria.

Metabolomics is routinely used in discovery studies to identify metabolic biomarkers of disease. This type of work requires a significant number of diseased and non-diseased subject samples collected using a specific blood collection tube. The type of blood collection tube used for biomarker discovery is relevant since analyte levels can be significantly altered by the tube

additive. Nevertheless, there may be instance when it is necessary to know whether a disease biomarker, discovered in one tube type, might be transferrable to a different tube type. A relevant example is kidney function biomarkers where serum is the preferred sample in the United States and lithium heparinized plasma is the preferred sample type in Europe. We used metabolomics to identify 15 serum metabolite biomarkers for estimation of glomerular filtration rate (GFR) [18]. Four (4) were selected to develop an algorithm for a more accurate estimation of GFR (i.e., compared to eGFR) [19]. The correlation between the biomarkers in serum versus lithium heparin plasma were strong suggesting that the biomarkers identified in serum might be transferrable to lithium heparinized plasma (Figs 3 and 4). When targeted assays for these four biomarkers, plus creatinine, were developed and tested in serum and lithium heparin plasma samples from ten subjects, $R^2$ values ranged from 0.94 (tryptophan) to 0.99 (acetyl-threonine) (S3 Table). We also evaluated the correlation of these five biomarkers in serum and lithium heparin plasma verses EDTA plasma and found strong correlation (S1 Table; serum versus EDTA plasma fold change between 0.95 to 1.04 and lithium heparin plasma verses EDTA plasma fold change between 0.96 to 1.02) suggesting that assessment of kidney function using these five biomarkers may be transferrable to EDTA plasma.

Limitations of this study should be considered. A myriad of factors such as diet, prandial state, gender, age, and genetic background can influence plasma levels of small molecule analytes [7, 8, 48–50]. These factors may have unanticipated effects not fully studied here but we controlled for fasting status. In a previous study, we showed elevated lactate levels and decreased arginine levels were strong biomarkers for plasma separation delays [9]. However, plasma lactate elevations can be precipitated by numerous other factors including intense exercise or mitochondrial disease [51]. Arginine levels can be highly skewed in critically ill individuals or in patients with argininemia- a rare inherited metabolic disease caused by a loss of arginase activity [52, 53]. In our study design we attempted to mitigate variances in analyte measurement by analyzing patient sets within the same analytical batch, obtaining samples within a small window (all samples acquired within one week), and all samples were obtained from patients at the same location using the same procedures. Nevertheless, all analytical methods, including metabolomics, have some level of intra-assay variability, as we have shown elsewhere [17]. Analyzing each sample in triplicate would have further mitigated intra-assay variability but would have also significantly increased the cost of this study and analyzing a larger sample set would increase the resolution and specificity of differences between the samples types.

## Conclusions

Our study demonstrates that different additives used in blood collection can affect the levels of numerous clinically relevant biochemicals; this is a significant challenge to assay and tube manufacturers and ultimately affects the analysis of biomarkers identified through clinical metabolomics studies. The optimal sample type needs to be utilized for both research and clinical studies and consistent collection methods ensure that pre-analytical error is minimized during assay development and clinical data collection. It is particularly relevant to the application of clinical metabolomics since pre-analytical errors resulting in the use of an inconsistent blood collection tube can result in significant alterations to metabolic pathways. Fortunately, it is possible to control for these pre-analytical errors in clinical metabolomics by actively surveying patient samples for evidence of inconsistent tube type. Jain et al. [9] described quality control measures for assessing pre-analytical error in sample preparation. Here, we describe the use of clinical metabolomics to ensure consistent blood collection tube submission. Future

efforts may include studies to identify instances of patient non-compliance (i.e., fasting status), extended storage, excessive freeze-thawing and implementation of delta checks to identify instances of mislabeled samples. Any pre-analytical change to a patient sample can have significant effects on individual analyte measurement and it is the comprehensive nature of this technology and its sensitivity to changes in conditions that make it ideal for assessing sample quality.

## Supporting information

**S1 Fig. Representative chromatograms for each of the methods utilized on the platform.**
A) HILIC Polar method, B) LC-MS/MS Negative, C) LC-MS/MS Positive Late, and D) LC-MS/MS Positive Early.
(TIF)

**S2 Fig. Hierarchical clustering of the five sample types.** CP—citrate plasma; HP—heparinized plasma; S—serum; EP—EDTA plasma; EPWB—EDTA anticoagulated whole blood. The three digit codes indicate the blinded donor number for the study.
(TIF)

**S1 Table.**
(XLSX)

**S2 Table. Summary of statistical differences between sample types.**
(DOCX)

**S3 Table. Correlation analysis for biochemicals across the 5 sample matrices.** Metabolites considered for analysis required at least 50% fill (50% of samples for that matrix type needed to have metabolite detected).
(XLSX)

## Author Contributions

**Conceptualization:** Adam D. Kennedy, Anne M. Evans, Douglas R. Toal.

**Data curation:** Matthew Mitchell.

**Formal analysis:** Adam D. Kennedy, Bryan Wittmann, Jesse Conner, Jacob Wulff, Matthew Mitchell.

**Investigation:** Lisa Ford, Bryan Wittmann, Jesse Conner.

**Methodology:** Lisa Ford, Jesse Conner, Anne M. Evans.

**Supervision:** Douglas R. Toal.

**Validation:** Matthew Mitchell.

**Writing – original draft:** Adam D. Kennedy.

**Writing – review & editing:** Adam D. Kennedy, Lisa Ford, Bryan Wittmann, Jesse Conner, Jacob Wulff, Matthew Mitchell, Anne M. Evans, Douglas R. Toal.

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
