## [Decision Letter · Decision Letter 0]

19 Feb 2021

PONE-D-20-34468

Global Biochemical Analysis of Plasma, Serum and Whole Blood Collected Using Various Anticoagulant Additives

PLOS ONE

Dear Dr. Adam Kennedy,

Thank you for submitting your manuscript to PLOS ONE. After careful consideration, we feel that it has merit but does not fully meet PLOS ONE’s publication criteria as it currently stands. Therefore, we invite you to submit a revised version of the manuscript that addresses the points raised during the review process.

We look forward to receiving your revised manuscript.

Kind regards,

Ch Ratnasekhar, Ph.D.

Academic Editor

PLOS ONE

Journal Requirements:

4.Thank you for stating the following in the Competing Interests:

"All authors are employees of Metabolon and, as such, have affiliations with or financial involvement with Metabolon, Inc.  "

We note that one or more of the authors have an affiliation to the commercial funders of this research study : Metabolon, Inc.

Reviewers' comments:

Reviewer's Responses to Questions

**Comments to the Author**

1. Is the manuscript technically sound, and do the data support the conclusions?

Reviewer #1: Yes

Reviewer #2: Yes

2. Has the statistical analysis been performed appropriately and rigorously? 

Reviewer #1: Yes

Reviewer #2: Yes

3. Have the authors made all data underlying the findings in their manuscript fully available?

Reviewer #1: No

Reviewer #2: Yes

4. Is the manuscript presented in an intelligible fashion and written in standard English?

Reviewer #1: Yes

Reviewer #2: Yes

5. Review Comments to the Author

Reviewer #1: The work in the manuscript entitled, “Global Biochemical Analysis of Plasma, Serum and Whole Blood Collected Using Various Anticoagulant Additives” is very interesting and have high significance. The work in the manuscript focused on importance of blood sampling using different anticoagulant additives for global metabolomics platform.

Some comments to improve the quality of the manuscript are as follows:

• In Introduction section, please provide more references.

• What was the storage conditions of the samples analysed? Did any experiment conducted to check the effect of sample degradation with respect to storage?

• What are the effects of these different anticoagulants on the enzymatic degradation of the metabolites?

• A reference can be seen in the manuscript for the extraction method used. However, this step has high significance on the proposed work. Hence, I would like to suggest to briefly mention about this step in the manuscript.

• What has been done for sample clean up prior injection to LC-MS analysis?

• As the method is for global metabolomics platform, sample size (n=27) could be a limitation of the study for statistical significance outcome.

• How many QCs were run in between the intra and inter samples batches?

• Validation of some key analytical parameters such as accuracy and precision must be provided in the manuscript.

• Please provide caption in the PCA figure or may be 2D PCA diagram with at least colour caption would be more suitable.

• Hierarchical clustering figure can be moved to supplementary file.

• Please provide some LC-MS chromatogram in the supplementary file.

• Please add a separate section about of limitation of the current study.

Reviewer #2: The manuscript “Global Biochemical Analysis of Plasma, Serum and Whole Blood Collected Using Various Anticoagulant Additives” explores different techniques involved in blood sample collection. The manuscript is well written, however some information must be included to avoid any technical bias

Could the authors include information that could affect the final outcome including food consumption and posture.

The authors have mentioned that 44% of subjects were females, please mention about menstrual cycle and pregnancy states as this could impact the final analysis

Was any of the subjects on prescription medication?

In order to avoid any technical bias, could the authors comment on the relative standard deviation between the samples as sample collection (27 subjects) and processing (135 samples) was done at different time points?

6. PLOS authors have the option to publish the peer review history of their article (what does this mean?). If published, this will include your full peer review and any attached files.

Reviewer #1: No

Reviewer #2: No

---

## [Author Response · Author response to Decision Letter 0]

9 Mar 2021

February 24, 2021

Dr. Ch Ratnasekhar

Academic Editor

PLoS ONE

Re: Resubmission of manuscript, PONE-D-20-34468

Dear Dr. Heber,

Thank you for the opportunity to resubmit our manuscript, Global Biochemical Analysis of Plasma, Serum and Whole Blood Collected Using Various Anticoagulant Additives. We look forward to the decision on our manuscript and future publication in The Journal. 

In addition to the significant amount of information that can be utilized and leveraged for developing and performing quality control of diagnostic tests of small molecules using mass spectrometry, we have addressed all comments by the reviewers point-by-point in the following pages. We are looking forward to the decision by The Journal. 

We hope that the manuscript will be sufficient and suitable for publication in the journal PLoS ONE.

We shall look forward to hearing from you after the review of the resubmission of our article.

Yours sincerely,

Adam D. Kennedy

Reviewers' comments:

Reviewer's Responses to Questions

Reviewer #1: The work in the manuscript entitled, “Global Biochemical Analysis of Plasma, Serum and Whole Blood Collected Using Various Anticoagulant Additives” is very interesting and have high significance. The work in the manuscript focused on importance of blood sampling using different anticoagulant additives for global metabolomics platform.

Some comments to improve the quality of the manuscript are as follows:

• In Introduction section, please provide more references.

Thank you for this comment. We have performed additional literature searches and incorporated addition references from the field. 

• What was the storage conditions of the samples analysed? Did any experiment conducted to check the effect of sample degradation with respect to storage?

All blood samples were drawn and plasma or serum separated within one hour of collection. Whole blood samples were aliquoted immediately and frozen after collection. Plasma and serum were frozen in 1 mL aliquots at -80oC post centrifugation and stored until analysis (1 month later). Previous studies have examined storage conditions. We know that long-term storage (2 years) at -80oC does not result in the degradation of biochemical profiles. The biggest contributors to sample degradation are processing delays (delays in separating plasma or serum from the other whole blood components) and freeze-thaw cycles. Both were avoided through stringent blood and sample collection protocols and creation of single-use aliquots for analysis. 

• What are the effects of these different anticoagulants on the enzymatic degradation of the metabolites?

We have expanded the first paragraph of our discussion to include this information. 

• A reference can be seen in the manuscript for the extraction method used. However, this step has high significance on the proposed work. Hence, I would like to suggest to briefly mention about this step in the manuscript.

Thank you for this comment. We have added additional information regarding the extraction of small molecules from samples into the methods section of the manuscript. 

• What has been done for sample clean up prior injection to LC-MS analysis?

We crash the proteins, divide the supernatant, dry under nitrogen, reconstitute in the respective buffers for each chromatographic analysis and run the samples. We assess carryover by injecting water blanks (process blanks) throughout the analysis to identify those molecules that are present in tubes and solvents. Any compound not present 3× above water blank levels are removed. 

• As the method is for global metabolomics platform, sample size (n=27) could be a limitation of the study for statistical significance outcome.

Thank you for this comment. We have added this to the limitations section in the Discussion of the manuscript. 

• How many QCs were run in between the intra and inter samples batches?

Within each sample batch, 2 process blanks and 6 quality control samples are run and analyzed. For this analysis, there were four sample batches meaning 8 process blanks and 24 quality control samples were run in total. The 6 quality control samples are a pool of plasma samples consisting of samples from >100 donors. Biochemical and analytical variability are assessed from these quality control samples and this analysis is utilized to give the RSD’s reported in the manuscript. 

• Validation of some key analytical parameters such as accuracy and precision must be provided in the manuscript.

We have performed a validation of our untargeted metabolomics platform and those results are published in a peer-reviewed journal (Ford et al. Journal of Applied Laboratory Medicine, 2019). We have expanded upon the information in this reference in both the introduction and discussion sections of the manuscript. 

• Please provide caption in the PCA figure or may be 2D PCA diagram with at least colour caption would be more suitable.

Thank you for this comment. We have adjusted the caption for the PCA figure in the manuscript.

• Hierarchical clustering figure can be moved to supplementary file.

Thank you for this comment. We have moved the hierarchical clustering figure to a supplementary figure file (Supplementary File 2). 

• Please provide some LC-MS chromatogram in the supplementary file.

We have supplied a chromatogram for the study and it is now the new Supplementary Figure 1. 

• Please add a separate section about of limitation of the current study.

Thank you for this comment. We have expanded paragraph in the Discussion outlining potential limitations for the study. 

Reviewer #2: The manuscript “Global Biochemical Analysis of Plasma, Serum and Whole Blood Collected Using Various Anticoagulant Additives” explores different techniques involved in blood sample collection. The manuscript is well written, however some information must be included to avoid any technical bias

Could the authors include information that could affect the final outcome including food consumption and posture.

Thank you for this comment. We have included this information in the limitations section of our manuscript. Food consumption, posture, and medications can affect the metabolomic profile of an individual. The strength of our manuscript was to profile the biochemical levels within each patient across the different blood collection anticoagulants. All subjects had fasted for at least 8 hours prior to giving their samples. All samples were acquired during the same blood draw for each patient. 

The authors have mentioned that 44% of subjects were females, please mention about menstrual cycle and pregnancy states as this could impact the final analysis

None of the subjects in the study were pregnant at the time of their blood draw. It was a requirement that no subjects were pregnant at the time of their draw. As with the question regarding medications below, the analyses performed in our statistical analyses were centered around matched-pair t-tests to identify differences from sample type to sample type WITHIN an individual. The menstrual cycle status of the subjects in the study is important, but not a central factor in the analysis. The metabolites that change with menstrual status would contribute to the range of detection in the study. We have added this information to the section outlining the donors and sample collection. 

Was any of the subjects on prescription medication?

Some of the patients were taking prescription medications and any medications that fit the profile of a small molecule that were detected in the study are listed in Supplementary table 1. Medications can alter biochemical profiles but the analyses performed in our statistical analyses were centered around matched-pair t-tests in order to identify differences from sample type to sample type WITHIN an individual. For many study designs, the medication status of the individuals in the study do need to be considered so that consistent baselines are achieved in order to assess the effect of those medicines on biomarker levels and indications linked to the mechanism of action of a drug. We have added this information to the section in the Materials and Methods outlining the donors and sample collection. 

In order to avoid any technical bias, could the authors comment on the relative standard deviation between the samples as sample collection (27 subjects) and processing (135 samples) was done at different time points?

All samples from a single individual were collected on the same day and during the same blood draw appointment. All individuals donated samples in the morning post >8hr fasting period. All samples were thawed, extracted and analyzed over the LC/MS platforms at a single timepoint in order to minimize any variation in sample preparation. The information in the Materials and Methods section describing the relative standard deviations (RSD) for the biochemicals is the analysis of the overall analytical variability in the system. The RSDs for the internal/labeled standards was 4% and the RSD for the endogenous biochemicals was 11%.

---

## [Editor Report · Decision Letter 1]

25 Mar 2021

Global Biochemical Analysis of Plasma, Serum and Whole Blood Collected Using Various Anticoagulant Additives

PONE-D-20-34468R1

Dear Dr. Adam,

We’re pleased to inform you that your manuscript has been judged scientifically suitable for publication and will be formally accepted for publication once it meets all outstanding technical requirements.

Kind regards,

Ch Ratnasekhar, Ph.D.

Academic Editor

PLOS ONE
---

## [Editor Report · Acceptance letter]

30 Mar 2021

PONE-D-20-34468R1 

Global Biochemical Analysis of Plasma, Serum and Whole Blood Collected Using Various Anticoagulant Additives 

Dear Dr. Kennedy:

I'm pleased to inform you that your manuscript has been deemed suitable for publication in PLOS ONE. Congratulations! Your manuscript is now with our production department. 

Kind regards, 

on behalf of

Dr. Ch Ratnasekhar 

Academic Editor

PLOS ONE